# CONTROLLING CHANGES TO ATTENTION LOGITS

## ABSTRACT

Stability of neural network weights is critical when training transformer models. The query and key weights are particularly problematic, as they tend to grow large without any intervention. Applying normalization to queries and keys, known as 'QK norm', fixes stability issues in practice, but is not always applicable. For example, QK norm is not compatible with Multi-head Latent Attention (MLA) because QK norm requires full materialization of queries and keys during inference, which is not done in MLA. In this paper we hypothesize that instability is driven primarily by changes in attention logits, rather than by their absolute magnitude, and that controlling these changes is sufficient for stability. We show that these changes are controllable by assigning parameter-dependent learning rates to the query and key weights. Our cheap intervention allows us to increase the base learning rate of the network, outperform other methods in the MLA setting, and achieve performance competitive with QK norm when using Multi-head Attention.

## 1 INTRODUCTION

Principled scaling of transformer models is crucial for efficiently training larger and more capable architectures. Maximal Update Parametrization ($\mu$P) (Yang et al., 2022) has emerged as a key technique in this area, enabling the transfer of optimal hyperparameters from smaller to larger models by carefully parameterizing the model. A core desideratum of $\mu$P is to control the magnitude of activations and their updates (Dey et al., 2025), ensuring consistent training dynamics across different model widths. Regarding attention, Yang et al. (2022) addresses attention logits blowing up as we increase model width by proposing a static attention scaling factor. While this static scaling helps control logit magnitude across different model widths, it does not address logit changes during longer training runs, which can become a major source of instability, particularly at high learning rates.

Attention logits are a well-known source of training instability (Zhai et al., 2023; Bai et al., 2025), and we illustrate this in Figure 1. Several interventions have emerged such as QK norm (Henry et al., 2020) and QK clip/MuonClip (Bai et al., 2025) to ensure their stability. While QK norm is especially effective, it is ill-suited for Multi-head Latent Attention (MLA) (Liu et al., 2024), as queries and keys are not fully materialized at inference-time for efficiency reasons (Bai et al., 2025). Other methods like QK clip require a bespoke attention mechanism to track maximum attention logits, which can complicate integration into existing codebases.

We hypothesize that instability of attention is driven mostly by large changes to logits, rather than by large logits themselves. This is motivated by the observation that the quadratic nature of attention means that gradients for the logits (and queries and keys) depend on the size of activations. Thus we can encounter unstable dynamics where large activations lead to large changes, which lead to large activations, etc.. We propose a simple method to resolve this, that modulates learning rates of query and key weight matrices, and we validate that this removes training instabilities. Specifically, our results demonstrate that our method is as stable as QK norm, particularly at high base learning rates. While not quite reaching the same peak performance as QK norm in the standard Multi-Head Attention (MHA) setting, our method is computationally cheaper and is applicable to MLA. When used with MLA our method enables higher base learning rates, and outperforms QK clip, highlighting potential benefits when training modern, efficient transformer architectures.

In summary, our contributions are as follows,

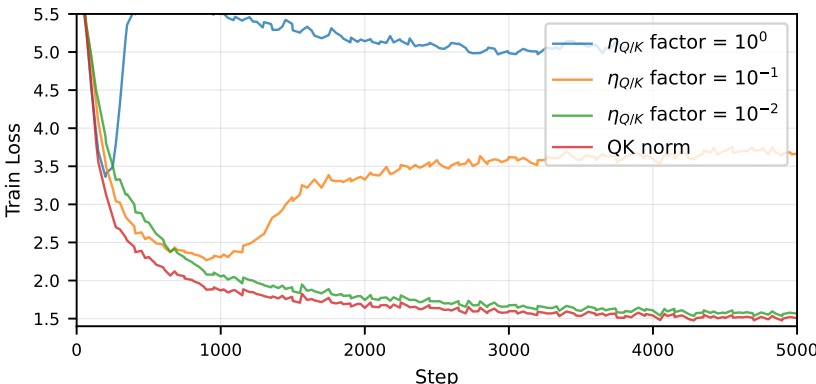

Figure 1: **Learning rate of query/key matrices is a critical factor for transformer pretraining stability.** 4 models are trained with a large base learning rate of $\eta = 3\mathrm{e} - 2$ for each parameter. Decreasing the learning rates of query and key weights ($\eta_{Q/K}$) alone, fully stabilizes pretraining. QK norm is shown to illustrate a stable baseline.

- We propose that changes to attention logits are a key quantity to track for stability in attention.
- We show that we can control logit changes by modulating the learning rate of query weights based on the norms of corresponding key weights, and vice versa, dynamically at training time.
- We demonstrate that this learning-rate intervention yields stable training and competitive validation loss, outperforming alternatives in the MLA setting.

## 2 RELATED WORK

**Controlling attention logits.** Training instabilities are often encountered in the attention layer itself. Attention logits may become large (Bai et al., 2025), potentially inducing collapse in attention entropy (Zhai et al., 2023), where attention distributions become highly concentrated. QK normalization (Henry et al., 2020), which applies normalization to query and key activations, has emerged as a simple and effective remedy, preventing large logits (Dehghani et al., 2023) and allowing larger learning rates (Wortsman et al., 2023). Similar methods such as logit soft-capping apply normalization to logits directly (Bello et al., 2016; Riviere et al., 2024). Other methods normalize the weights rather than activations: $\sigma$Reparam (Zhai et al., 2023) parameterizes weights into a matrix and a scalar component, with the matrix having unit spectral norm and a scalar that captures overall scale; QK clip (Bai et al., 2025) controls attention logits by clipping weights whenever the logits grow beyond a certain threshold.

**Parameter-specific learning rates.** While it is common to share the same learning rate across all parameters in a neural network, parameter-specific learning rates have been extensively examined (Milsom et al., 2025; You et al., 2017; Liu et al., 2019; Xu et al., 2019; Wang et al., 2025; Bernstein et al., 2020; Qi et al., 2025; Yang et al., 2023). Proposals often include adjusting the learning rate of a parameter according to the norm of step/gradient (Yang et al., 2023; Liu et al., 2019), as well as the parameter itself (Qi et al., 2025), such as LARS, LAMB, and Fromage (Bernstein et al., 2020; You et al., 2017; 2019).

Our work selects parameter-specific learning rates that control changes to attention logits. However, by considering attention logits as a whole, our parameter-specific learning rates are 'inter-parameter', unlike other methods, such as LARS, which consider each parameter tensor independently. Our method is also inspired by $\mu$P (Yang et al., 2022; Dey et al., 2025); in $\mu$P, one of the desiderata is that as we make changes to our parameters in a network, the residual stream should correspondingly change in a controlled, 'order 1-like' manner. Our work extends this notion to logits.

---

**Algorithm 1** QuacK (MHA)

---

**Require:** Hyperparameter $\tau$, base learning rate $\eta$
  Make the following additions to the transformer training script:

  # At initialization. Calculate initial norms for query/key weights for all heads
  **for all** layers $\ell$ **do**
    **for all** heads $h$ **do**
      $\mathbf{W}_Q^{\ell,h}.\texttt{init\_norm} \leftarrow \|\mathbf{W}_Q^{\ell,h}\|$
      $\mathbf{W}_K^{\ell,h}.\texttt{init\_norm} \leftarrow \|\mathbf{W}_K^{\ell,h}\|$
    **end for**
  **end for**

  # During training. Prior to each optimization step, adjust learning rates
  **for all** layers $\ell$ **do**
    **for all** heads $h$ **do**
      $\mathbf{W}_Q^{\ell,h}.\texttt{lr} \leftarrow \tau\,\eta \cdot \dfrac{\mathbf{W}_K^{\ell,h}.\texttt{init\_norm}}{\|\mathbf{W}_K^{\ell,h}\|}$
      $\mathbf{W}_K^{\ell,h}.\texttt{lr} \leftarrow \tau\,\eta \cdot \dfrac{\mathbf{W}_Q^{\ell,h}.\texttt{init\_norm}}{\|\mathbf{W}_Q^{\ell,h}\|}$
    **end for**
  **end for**

---

## 3 METHODS

Unlike other transformer modules, attention has quadratic structure. In particular, the attention logits are given by,

$$\mathbf{L} = d^{-1/2}\mathbf{Q}\mathbf{K}^T = d^{-1/2}\mathbf{X}\mathbf{W}_Q\mathbf{W}_K^T\mathbf{X}^T. \tag{1}$$

We attempt to keep the changes to attention logits, $\Delta\mathbf{L}$, under control. By a first-order analysis, we see that if the queries are large, then perturbations due to the keys will be amplified, and vice versa:

$$\Delta\mathbf{L} = \frac{(\mathbf{Q}+\Delta\mathbf{Q})(\mathbf{K}+\Delta\mathbf{K})^T - \mathbf{Q}\mathbf{K}^T}{\sqrt{d}} \approx \frac{\mathbf{Q}(\Delta\mathbf{K})^T + (\Delta\mathbf{Q})\mathbf{K}^T}{\sqrt{d}}. \tag{2}$$

The main tool we have for controlling changes is the learning rate. Thus we propose to set the learning rates $\eta_Q$, $\eta_K$ (for $\mathbf{W}_Q$, and $\mathbf{W}_K$ respectively) such that $\mathbf{Q}(\Delta\mathbf{K})^T$ and $(\Delta\mathbf{Q})\mathbf{K}^T$ are both 'order 1'. We formalize this notion in Lemma 1.

Following the Lemma we set,

$$\eta_Q \propto \|\mathbf{W}_K\|^{-1}, \ \ \eta_K \propto \|\mathbf{W}_Q\|^{-1}. \tag{3}$$

In practice we treat the constant of proportionality in Eq. (3) as a hyperparameter: at initialization, we set the learning rate for each query and key weight to be equal to $\tau\eta$ and we tune $\tau$. Thus $\tau$ acts as a relative initial learning rate (relative to $\eta$, the base learning rate). This allows for clear comparison to other methods in the experiments.

The above methodology applies to both the single- and multi-head (MHA) setting. In MHA, each head has its own query and key weight, so we apply Eq. (3) to each head separately. We summarize the resulting method in Algorithm 1.

We extend to MLA using a similar approach in Appendix C. A notable difference between MHA and MLA is that there are several more parameter matrices to consider; bounding the change in logits requires us to adjust the learning rate of each of these parameters. We detail exactly how to set the learning rates for MLA in Algorithm 2.

## 4 EXPERIMENTS

To evaluate the different approaches, we trained $\sim$1B-sized transformer LMs (Qwen3-based) with both MHA (Vaswani et al., 2017) and MLA (Liu et al., 2024). Runs used the same optimizer

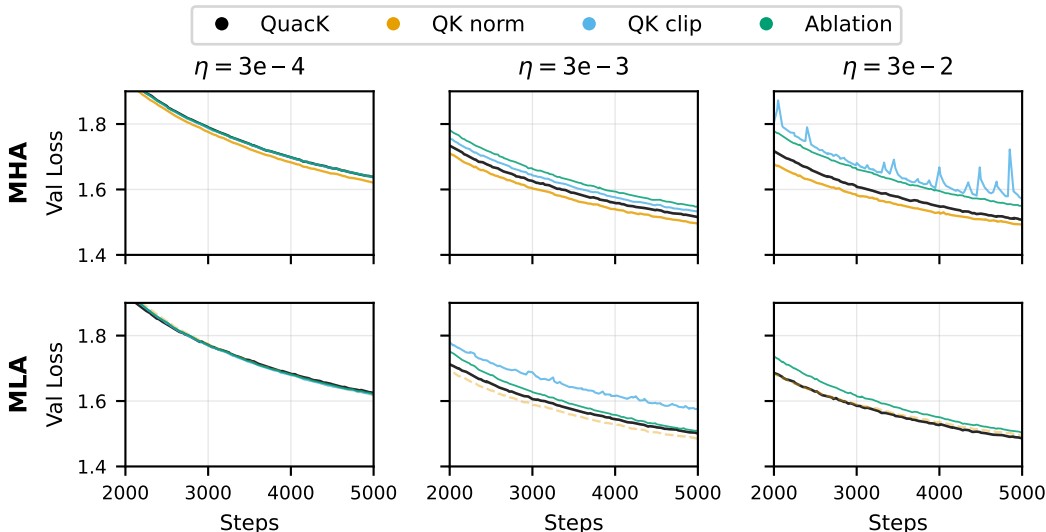

Figure 2: Validation losses when training each method with `attn` ∈ {`MHA`, `MLA`}, and learning rates, $\eta \in \{3e-4, 3e-3, 3e-2\}$. QK clip is unstable at high learning rates (omitted from the bottom right plot due to loss $\gg 2$). QK norm is overall the most performant, but it is not appropriate for use with MLA at inference-time for efficiency reasons (illustrated via dashed yellow line in the MLA row). QuacK is a sensible alternative, as it is stable in the high LR setting, performant, and applicable in the MLA setting.

(Muon (Jordan et al., 2024)) and data pipeline (Cosmopedia-V2 / SmolLM-corpus (Ben Allal et al., 2024)); details of model hyperparameters and compute setup are provided in the Appendix.

Our experiments varied attention type (`MHA` / `MLA`), base learning rate $\eta \in \{3e-4, 3e-3, 3e-2\}$, and the stabilization method: QK norm (Henry et al., 2020); QK clip (Bai et al., 2025); fixed LR scaling for Q/K weights (ablation), and QuacK (ours).

**Higher learning rates are better.** Figure 2, left column, shows that at the low learning rate of $\eta = 3e-4$, all logit interventions perform similarly, but with QK norm performing marginally better. The lack of variety in performance is likely due to the fact the learning rate is small enough that we don't encounter instabilities. However, performance is much improved by increasing the learning rate (column 2, 3).

**QuacK maintains stability and strong performance, enabling higher base learning rates.** QK clip is insufficient to prevent instabilities at the highest base learning rate of $\eta = 3e-2$ (column 3, Figure 2), and it underperforms, especially in the MLA setting, when $\eta = 3e-3$ (column 2). The ablation, which sets the learning rates for query and key weights to smaller fixed values, is stable, but underperforms QuacK in both the MHA and MLA settings. QuacK has similar but slightly worse performance compared to QK norm in the MHA setting, but is the best performing method in the MLA setting (we include QK norm results in the MLA setting for comparison, but it is not viable at inference-time like the other methods).

## 5 CONCLUSION

Our method, QuacK, stabilizes training by controlling changes in attention logits via coupled learning rates for query/key weights. In ~1B-scale pretraining, QuacK enables higher base learning rates and improves performance over QK clip in the MLA setting (details and additional plots in the appendix).

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

## A    EXPERIMENTAL SETUP DETAILS

All models used $d_{\text{model}} = 2048$, $d_{\text{ff}} = 4d_{\text{model}}$, $n_{\text{head}} = 32$, $n_{\text{layer}} = 14$, and were trained for 5000 steps at context length 2048 with 96 sequences per batch. We used the GPT-2 tokenizer (Radford et al., 2019) (vocab size 49,152) with embedding/unembedding weight tying, and trained on Cosmopedia-V2 / SmolLM-corpus (Ben Allal et al., 2024) using Muon (Jordan et al., 2024) with constant LR and 500 warmup steps. MHA used $d_{\text{head}} = 64$; MLA used $d_{\text{head}} = 128$ (with $d_{\text{rope}} = d_{\text{nope}} = 64$), latent dims $d_{\text{cq}} = 512$ and $d_{\text{ckv}} = 256$.

**Lemma 1.** *Let* $\mathbf{W}_Q, \mathbf{W}_K \in \mathbb{R}^{d_{model} \times d_{head}}$ *be weight matrices corresponding to a particular attention head, and consider the worst-case change in logits, for unit normed input,*

$$\max_{\|x\|_2 = \|y\|_2 = 1} |\Delta\ell| := \max_{\|x\|_2 = \|y\|_2 = 1} |x^\top (\mathbf{W} + \Delta\mathbf{W})y - x^\top \mathbf{W}y|,$$

*where* $\mathbf{W} = d_{head}^{-1/2} \mathbf{W}_Q^\top \mathbf{W}_K$. *Suppose that the steps for* $\mathbf{W}_Q$ *and* $\mathbf{W}_K$ *are given by* $\Delta\mathbf{W}_{Q/K} = -\eta_{Q/K}\mathbf{G}_{Q/K}$, *where* $\|\mathbf{G}_{Q/K}\| \leq D$ *for some constant* $D$ *(which is the case for Adam and Muon). If there is a constant* $c$ *such that* $0 < c \leq \|\mathbf{W}_Q\|, \|\mathbf{W}_K\|$, *and the learning rates satisfy* $\eta_Q \propto \|\mathbf{W}_K\|^{-1}$, *and* $\eta_K \propto \|\mathbf{W}_Q\|^{-1}$, *then the worst-case change in logits is bounded above independently of the norm of the weights.*

## B    PROOF OF LEMMA 1

**Lemma 1.** *Let* $\mathbf{W}_Q, \mathbf{W}_K \in \mathbb{R}^{d_{model} \times d_{head}}$ *be weight matrices corresponding to a particular attention head, and consider the worst-case change in logits, for unit normed input,*

$$\max_{\|x\|_2 = \|y\|_2 = 1} |\Delta\ell| := \max_{\|x\|_2 = \|y\|_2 = 1} |x^\top (\mathbf{W} + \Delta\mathbf{W})y - x^\top \mathbf{W}y|,$$

*where* $\mathbf{W} = d_{head}^{-1/2} \mathbf{W}_Q^\top \mathbf{W}_K$. *Suppose that the steps for* $\mathbf{W}_Q$ *and* $\mathbf{W}_K$ *are given by* $\Delta\mathbf{W}_{Q/K} = -\eta_{Q/K}\mathbf{G}_{Q/K}$, *where* $\|\mathbf{G}_{Q/K}\| \leq D$ *for some constant* $D$ *(which is the case for Adam and*

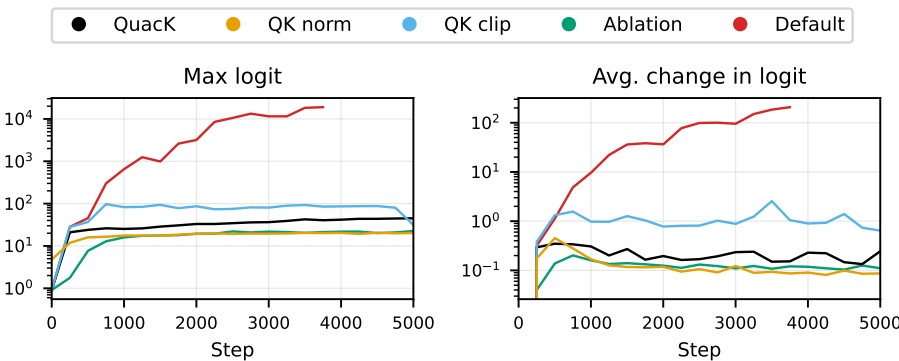

Figure 3: Max logit (left) and average absolute change in logit throughout training (right) with a base learning rate of $\eta = 3\mathrm{e} - 3$. Here we show the middle head of the middle layer (head 16 and layer 8) while training with MLA.

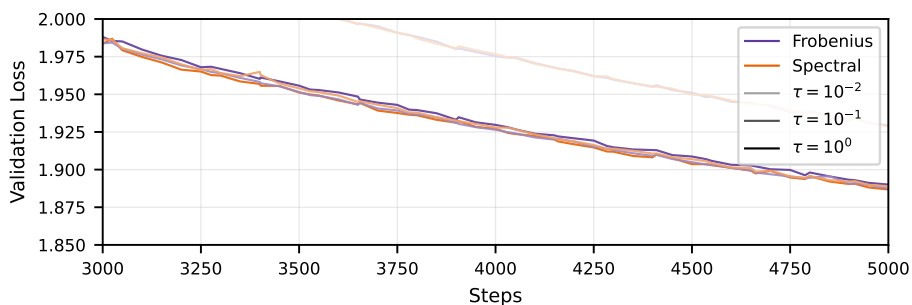

Figure 4: Performance differences when applying Algorithm 1 with different norms are small. We show validation losses when training a small model ($\sim 100$M parameters) with Algorithm 1 to modulate the query and key weight learning rates. Different curves show results with different values of the hyperparameter $\tau$ and measuring the query and key weights with either Frobenius or spectral norm.

*Muon). If there is a constant $c$ such that $0 < c \leq \|\mathbf{W}_Q\|, \|\mathbf{W}_K\|$, and the learning rates satisfy $\eta_Q \propto \|\mathbf{W}_K\|^{-1}$, and $\eta_K \propto \|\mathbf{W}_Q\|^{-1}$, then the worst-case change in logits is bounded above independently of the norm of the weights.*

*Proof.* The change in logits is given by,

$$d_{\mathrm{head}}^{1/2}|\Delta\ell| = |(q + \Delta q)^T(k + \Delta k) - q^T k|$$
$$= |(\Delta q)^T k + q^T \Delta k + (\Delta q)^T \Delta k|$$
$$\leq |(\Delta q)^T k| + |q^T \Delta k| + \|\Delta q\|\|\Delta k\|. \tag{4}$$

where,

$$q = \mathbf{W}_Q x, \ k = \mathbf{W}_K y \tag{5}$$

The query and key perturbations are given by,

$$\Delta q = (\mathbf{W}_Q + \Delta\mathbf{W}_Q)x - \mathbf{W}_Q x = \Delta\mathbf{W}_Q x, \tag{6}$$
$$\Delta k = (\mathbf{W}_K + \Delta\mathbf{W}_K)y - \mathbf{W}_K y = \Delta\mathbf{W}_K y. \tag{7}$$

We now bound the first order terms in Eq. (4), assuming inputs are unit normed,

$$|(\Delta q)^T k| = |(\Delta \mathbf{W}_Q x)^T (\mathbf{W}_K y)| = |x^T \Delta \mathbf{W}_Q^T \mathbf{W}_K y|$$

$$\leq \|x\| \|\Delta \mathbf{W}_Q^T \mathbf{W}_K\| \|y\|$$

$$\leq \eta_Q D \|\mathbf{W}_K\|, \tag{8a}$$

$$|q^T \Delta k| = |(\mathbf{W}_Q x)^T (\Delta \mathbf{W}_K y)| = |x^T \mathbf{W}_Q^T \Delta \mathbf{W}_K y|$$

$$\leq \|x\| \|\mathbf{W}_Q^T \Delta \mathbf{W}_K\| \|y\|$$

$$\leq \eta_K D \|\mathbf{W}_Q\|. \tag{8b}$$

For some constants $\tau_Q, \tau_K$, set,

$$\eta_Q = \tau_Q \|\mathbf{W}_K\|^{-1} \tag{9a}$$

$$\eta_K = \tau_K \|\mathbf{W}_Q\|^{-1}. \tag{9b}$$

Substituting these into Eqs. (8), we obtain the bounds,

$$|(\Delta q)^T k| \leq (\tau_Q \|\mathbf{W}_K\|^{-1}) D \|\mathbf{W}_K\| = \tau_Q D, \tag{10}$$

$$|q^T \Delta k| \leq (\tau_K \|\mathbf{W}_Q\|^{-1}) D \|\mathbf{W}_Q\| = \tau_K D. \tag{11}$$

Note that even if RoPE is applied, such that $q = R_x \mathbf{W}_Q x$, the bound remains identical as $\|R\mathbf{W}\| = \|\mathbf{W}\|$ (if the Frobenius or spectral is used).

Finally, we consider the quadratic term $\|\Delta q\| \|\Delta k\|$,

$$\|\Delta q\| \|\Delta k\| \leq \|\Delta \mathbf{W}_Q\| \|\Delta \mathbf{W}_K\|$$

$$= \frac{\tau_Q \tau_K \|\mathbf{G}_Q\| \|\mathbf{G}_K\|}{\|\mathbf{W}_Q\| \|\mathbf{W}_K\|}$$

$$\leq \frac{\tau_Q \tau_K D^2}{c^2}. \tag{12}$$

Thus the change in logits is bounded by a constant. □

## C  EXTENSION TO MLA

In this section we motivate Algorithm 2, specifically the factors associated with each weight.

We use a similar approach to Section 3 / Appendix B when extending to MLA (Ji et al., 2025; Liu et al., 2024). For now, assume the single-head setting. MLA tells us to calculate queries and keys as follows,

$$q = \text{Concat}(q_{\text{nope}}, q_{\text{rope}}) \tag{13a}$$

$$k = \text{Concat}(k_{\text{nope}}, k_{\text{rope}}) \tag{13b}$$

$$q_{\text{nope}} = \mathbf{W}_{\text{uq}} \mathbf{W}_{\text{dq}} x \tag{13c}$$

$$q_{\text{rope}} = R_x (\mathbf{W}_{\text{qr}} \mathbf{W}_{\text{dq}} x) \tag{13d}$$

$$c_{\text{kv}} = \mathbf{W}_{\text{dkv}} y \tag{13e}$$

$$k_{\text{nope}} = \mathbf{W}_{\text{uk}} c_{\text{kv}} = \mathbf{W}_{\text{uk}} \mathbf{W}_{\text{dkv}} y \tag{13f}$$

$$k_{\text{rope}} = R_y (\mathbf{W}_{\text{kr}} y). \tag{13g}$$

Here, $x$ and $y$ are two token embeddings. The 'down' matrices, $\mathbf{W}_{\text{dq}}$ and $\mathbf{W}_{\text{dkv}}$, project queries and keys/values respectively down to a lower dimensional latent space. This enables efficient caching of $c_{\text{kv}}$. The 'up' matrices $\mathbf{W}_{\text{uq}}, \mathbf{W}_{\text{uk}}$ project these latents up to a higher dimensional space for attention calculations on each head. The $\mathbf{W}_{\text{qr}}$ and $\mathbf{W}_{\text{kr}}$ matrices are used to produce decoupled queries and keys for RoPE (Su et al., 2021) embeddings, with the position embedding applied via the rotation matrices $R_x$ and $R_y$.

The change in logits is given by,

$$d_{\text{head}}^{1/2} |\Delta \ell| = |(q + \Delta q)^T (k + \Delta k) - q^T k| = |(\Delta q)^T k + q^T \Delta k + (\Delta q)^T \Delta k|, \tag{14}$$

---

**Algorithm 2** QuacK (MLA)

---

**Require:** Hyperparameter $\tau$, base learning rate $\eta$

   Make the following additions to the transformer training script:

   **function** `compute_lr_factors()`

      **for all** layers $\ell$ **do**

         **for all** heads $h$ **do**

            $\mathbf{W}_{\mathrm{uq}}^{\ell,h}$`.factor` $\leftarrow (\|\mathbf{W}_{\mathrm{dq}}^{\ell}\|\,\|\mathbf{W}_{\mathrm{uk}}^{\ell,h}\|\,\|\mathbf{W}_{\mathrm{dkv}}^{\ell}\|)^{-1}$

            $\mathbf{W}_{\mathrm{uk}}^{\ell,h}$`.factor` $\leftarrow (\|\mathbf{W}_{\mathrm{uq}}^{\ell,h}\|\,\|\mathbf{W}_{\mathrm{dq}}^{\ell}\|\,\|\mathbf{W}_{\mathrm{dkv}}^{\ell}\|)^{-1}$

            $\mathbf{W}_{\mathrm{qr}}^{\ell,h}$`.factor` $\leftarrow (\|\mathbf{W}_{\mathrm{dq}}^{\ell}\|\,\|\mathbf{W}_{\mathrm{kr}}^{\ell}\|)^{-1}$

         **end for**

      $\mathbf{W}_{\mathrm{dq}}^{\ell}$`.factor` $\leftarrow \min\Big\{(\max_h \|\mathbf{W}_{\mathrm{uq}}^{\ell,h}\|\|\mathbf{W}_{\mathrm{uk}}^{\ell,h}\|\|\mathbf{W}_{\mathrm{dkv}}^{\ell}\|)^{-1},\ (\max_h \|\mathbf{W}_{\mathrm{qr}}^{\ell,h}\|\|\mathbf{W}_{\mathrm{kr}}^{\ell}\|)^{-1}\Big\}$

      $\mathbf{W}_{\mathrm{dkv}}^{\ell}$`.factor` $\leftarrow (\max_h \|\mathbf{W}_{\mathrm{uq}}^{\ell,h}\|\|\mathbf{W}_{\mathrm{dq}}^{\ell}\|\|\mathbf{W}_{\mathrm{uk}}^{\ell,h}\|)^{-1}$

      $\mathbf{W}_{\mathrm{kr}}^{\ell}$`.factor` $\leftarrow (\max_h \|\mathbf{W}_{\mathrm{qr}}^{\ell,h}\|\|\mathbf{W}_{\mathrm{dq}}^{\ell}\|)^{-1}$

      **end for**

   **end function**

   $\{$`attention_weights`$\} \leftarrow \{\mathbf{W}_{\mathrm{uq}}^{\ell,h}, \mathbf{W}_{\mathrm{uk}}^{\ell,h}, \mathbf{W}_{\mathrm{qr}}^{\ell,h}, \mathbf{W}_{\mathrm{dq}}^{\ell}, \mathbf{W}_{\mathrm{dkv}}^{\ell}, \mathbf{W}_{\mathrm{kr}}^{\ell}$ for all layers $\ell$ for all heads $h\}$

   # At initialization. Compute initial learning rate factors for all attention weights

   `compute_lr_factors()`

   **for all** $\mathbf{W}$ in $\{$`attention_weights`$\}$ **do**

      $\mathbf{W}$`.init_factor` $\leftarrow \mathbf{W}$`.factor`

   **end for**

   # During training. Prior to each optimization step, adjust learning rates

   `compute_lr_factors()`

   **for all** $\mathbf{W}$ in $\{$`attention_weights`$\}$ **do**

      $\mathbf{W}$`.lr` $\leftarrow \tau\,\eta \cdot \dfrac{\mathbf{W}\text{`.factor`}}{\mathbf{W}\text{`.init_factor`}}$

   **end for**

---

and we can bound the change,

$$
\begin{aligned}
d_{\mathrm{head}}^{1/2}|\Delta\ell| &\leq |(\Delta q)^T k| + |q^T \Delta k| + \|\Delta q\|\|\Delta k\| \\
&\leq |(\Delta q_{\mathrm{nope}})^T k_{\mathrm{nope}}| + |(\Delta q_{\mathrm{rope}})^T k_{\mathrm{rope}}| + |q_{\mathrm{nope}}^T \Delta k_{\mathrm{nope}}| + |q_{\mathrm{rope}}^T \Delta k_{\mathrm{rope}}| + \|\Delta q\|\|\Delta k\|.
\end{aligned}
\tag{15}
$$

Expanding further, for the queries, we have,

$$
\begin{aligned}
\Delta q_{\mathrm{nope}} &= (\mathbf{W}_{\mathrm{uq}} + \Delta\mathbf{W}_{\mathrm{uq}})(\mathbf{W}_{\mathrm{dq}} + \Delta\mathbf{W}_{\mathrm{dq}})x - \mathbf{W}_{\mathrm{uq}}\mathbf{W}_{\mathrm{dq}}x \\
&= \Delta\mathbf{W}_{\mathrm{uq}}\mathbf{W}_{\mathrm{dq}}x + \mathbf{W}_{\mathrm{uq}}\Delta\mathbf{W}_{\mathrm{dq}}x + \Delta\mathbf{W}_{\mathrm{uq}}\Delta\mathbf{W}_{\mathrm{dq}}x,
\end{aligned}
\tag{16a}
$$

$$
\begin{aligned}
\Delta q_{\mathrm{rope}} &= R_x[(\mathbf{W}_{\mathrm{qr}} + \Delta\mathbf{W}_{\mathrm{qr}})(\mathbf{W}_{\mathrm{dq}} + \Delta\mathbf{W}_{\mathrm{dq}})x - \mathbf{W}_{\mathrm{qr}}\mathbf{W}_{\mathrm{dq}}x] \\
&= R_x[\Delta\mathbf{W}_{\mathrm{qr}}\mathbf{W}_{\mathrm{dq}}x + \mathbf{W}_{\mathrm{qr}}\Delta\mathbf{W}_{\mathrm{dq}}x + \Delta\mathbf{W}_{\mathrm{qr}}\Delta\mathbf{W}_{\mathrm{dq}}x],
\end{aligned}
\tag{16b}
$$

and for the keys,

$$
\begin{aligned}
\Delta k_{\mathrm{nope}} &= (\mathbf{W}_{\mathrm{uk}} + \Delta\mathbf{W}_{\mathrm{uk}})(\mathbf{W}_{\mathrm{dkv}} + \Delta\mathbf{W}_{\mathrm{dkv}})y - \mathbf{W}_{\mathrm{uk}}\mathbf{W}_{\mathrm{dkv}}y \\
&= \Delta\mathbf{W}_{\mathrm{uk}}\mathbf{W}_{\mathrm{dkv}}y + \mathbf{W}_{\mathrm{uk}}\Delta\mathbf{W}_{\mathrm{dkv}}y + \Delta\mathbf{W}_{\mathrm{uk}}\Delta\mathbf{W}_{\mathrm{dkv}}y
\end{aligned}
\tag{17a}
$$

$$
\Delta k_{\mathrm{rope}} = R_y[(\mathbf{W}_{\mathrm{kr}} + \Delta\mathbf{W}_{\mathrm{kr}})y - \mathbf{W}_{\mathrm{kr}}y] = R_y\Delta\mathbf{W}_{\mathrm{kr}}y.
\tag{17b}
$$

We now use these expressions, and the expressions for $q_{\mathrm{nope}}, k_{\mathrm{nope}}, q_{\mathrm{rope}}, k_{\mathrm{rope}}$, to bound each of the terms in Eq. (15). We will make some assumptions (similar to Lemma 1),

- the inputs $x$ and $y$ are unit normed;
- we use a submultiplicative norm (e.g. the Frobenius or Spectral norm);
- conditioned gradients are bounded by a constant, i.e. $\Delta\mathbf{W}_{\mathrm{x}} = -\eta_{\mathrm{x}}\mathbf{G}_{\mathrm{x}}$ where $\|\mathbf{G}_{\mathrm{x}}\| \leq D$ (valid for Muon and Adam);
- the weight norms are lower bounded by a constant $c$.

We consider the first order terms. We have,

$$
\begin{aligned}
|\Delta q_{\mathrm{nope}}^T k_{\mathrm{nope}}| &= |(\Delta \mathbf{W}_{\mathrm{uq}} \mathbf{W}_{\mathrm{dq}} x + \mathbf{W}_{\mathrm{uq}} \Delta \mathbf{W}_{\mathrm{dq}} x + \Delta \mathbf{W}_{\mathrm{uq}} \Delta \mathbf{W}_{\mathrm{dq}} x)^T k_{\mathrm{nope}}| \\
&\leq \|\Delta \mathbf{W}_{\mathrm{uq}}\| \|\mathbf{W}_{\mathrm{dq}}\| \|k_{\mathrm{nope}}\| + \|\mathbf{W}_{\mathrm{uq}}\| \|\Delta \mathbf{W}_{\mathrm{dq}}\| \|k_{\mathrm{nope}}\| + \|\Delta \mathbf{W}_{\mathrm{uq}}\| \|\Delta \mathbf{W}_{\mathrm{dq}}\| \|k_{\mathrm{nope}}\| \\
&\leq \eta_{\mathrm{uq}} D \|\mathbf{W}_{\mathrm{dq}}\| \|\mathbf{W}_{\mathrm{uk}}\| \|\mathbf{W}_{\mathrm{dkv}}\| + \eta_{\mathrm{dq}} D \|\mathbf{W}_{\mathrm{uq}}\| \|\mathbf{W}_{\mathrm{uk}}\| \|\mathbf{W}_{\mathrm{dkv}}\| + O(\eta_{\mathrm{uq}} \eta_{\mathrm{dq}} \|\mathbf{W}_{\mathrm{uk}}\| \|\mathbf{W}_{\mathrm{dkv}}\|)
\end{aligned}
\tag{18a}
$$

$$
\begin{aligned}
|(\Delta q_{\mathrm{rope}})^T k_{\mathrm{rope}}| &= |(R_x [\Delta \mathbf{W}_{\mathrm{qr}} \mathbf{W}_{\mathrm{dq}} x + \mathbf{W}_{\mathrm{qr}} \Delta \mathbf{W}_{\mathrm{dq}} x + \Delta \mathbf{W}_{\mathrm{qr}} \Delta \mathbf{W}_{\mathrm{dq}} x])^T k_{\mathrm{rope}}| \\
&\leq (\|\Delta \mathbf{W}_{\mathrm{qr}}\| \|\mathbf{W}_{\mathrm{dq}}\| + \|\mathbf{W}_{\mathrm{qr}}\| \|\Delta \mathbf{W}_{\mathrm{dq}}\| + \|\Delta \mathbf{W}_{\mathrm{qr}}\| \|\Delta \mathbf{W}_{\mathrm{dq}}\|) \|k_{\mathrm{rope}}\| \\
&\leq \eta_{\mathrm{qr}} D \|\mathbf{W}_{\mathrm{dq}}\| \|\mathbf{W}_{\mathrm{kr}}\| + \eta_{\mathrm{dq}} D \|\mathbf{W}_{\mathrm{qr}}\| \|\mathbf{W}_{\mathrm{kr}}\| + O(\eta_{\mathrm{dq}} \eta_{\mathrm{qr}} \|\mathbf{W}_{\mathrm{kr}}\|),
\end{aligned}
\tag{18b}
$$

$$
\begin{aligned}
|q_{\mathrm{nope}}^T \Delta k_{\mathrm{nope}}| &= |q_{\mathrm{nope}}^T (\Delta \mathbf{W}_{\mathrm{uk}} \mathbf{W}_{\mathrm{dkv}} y + \mathbf{W}_{\mathrm{uk}} \Delta \mathbf{W}_{\mathrm{dkv}} y + \Delta \mathbf{W}_{\mathrm{uk}} \Delta \mathbf{W}_{\mathrm{dkv}} y)| \\
&\leq \|q_{\mathrm{nope}}\| \|\Delta \mathbf{W}_{\mathrm{uk}}\| \|\mathbf{W}_{\mathrm{dkv}}\| + \|q_{\mathrm{nope}}\| \|\mathbf{W}_{\mathrm{uk}}\| \|\Delta \mathbf{W}_{\mathrm{dkv}}\| + \|q_{\mathrm{nope}}\| \|\Delta \mathbf{W}_{\mathrm{uk}}\| \|\Delta \mathbf{W}_{\mathrm{dkv}}\| \\
&\leq \eta_{\mathrm{uk}} D \|\mathbf{W}_{\mathrm{uq}}\| \|\mathbf{W}_{\mathrm{dq}}\| \|\mathbf{W}_{\mathrm{dkv}}\| + \eta_{\mathrm{dkv}} D \|\mathbf{W}_{\mathrm{uq}}\| \|\mathbf{W}_{\mathrm{dq}}\| \|\mathbf{W}_{\mathrm{uk}}\| + O(\eta_{\mathrm{uk}} \eta_{\mathrm{dkv}} \|\mathbf{W}_{\mathrm{uq}}\| \|\mathbf{W}_{\mathrm{dq}}\|),
\end{aligned}
\tag{18c}
$$

$$
|q_{\mathrm{rope}}^T \Delta k_{\mathrm{rope}}| = |q_{\mathrm{rope}}^T (R_y \Delta \mathbf{W}_{\mathrm{kr}} y)| \leq \|q_{\mathrm{rope}}\| \|\Delta \mathbf{W}_{\mathrm{kr}}\| \leq \eta_{\mathrm{kr}} D \|\mathbf{W}_{\mathrm{qr}}\| \|\mathbf{W}_{\mathrm{dq}}\|.
\tag{18d}
$$

We used the fact that for rotation matrices $R$, $\|\mathbf{W} R\| = \|R \mathbf{W}\| = \|\mathbf{W}\|$.

Ultimately, Eqs. (18) suggest to set the learning rates for each attention weight parameter as,

$$
\eta_{\mathrm{uq}} = \tau (\|\mathbf{W}_{\mathrm{dq}}\| \|\mathbf{W}_{\mathrm{uk}}\| \|\mathbf{W}_{\mathrm{dkv}}\|)^{-1}
\tag{19a}
$$

$$
\eta_{\mathrm{dq}} = \tau \min \left\{ (\|\mathbf{W}_{\mathrm{uq}}\| \|\mathbf{W}_{\mathrm{uk}}\| \|\mathbf{W}_{\mathrm{dkv}}\|)^{-1}, (\|\mathbf{W}_{\mathrm{qr}}\| \|\mathbf{W}_{\mathrm{kr}}\|)^{-1} \right\}
\tag{19b}
$$

$$
\eta_{\mathrm{qr}} = \tau (\|\mathbf{W}_{\mathrm{dq}}\| \|\mathbf{W}_{\mathrm{kr}}\|)^{-1}
\tag{19c}
$$

$$
\eta_{\mathrm{uk}} = \tau (\|\mathbf{W}_{\mathrm{uq}}\| \|\mathbf{W}_{\mathrm{dq}}\| \|\mathbf{W}_{\mathrm{dkv}}\|)^{-1}
\tag{19d}
$$

$$
\eta_{\mathrm{dkv}} = \tau (\|\mathbf{W}_{\mathrm{uq}}\| \|\mathbf{W}_{\mathrm{dq}}\| \|\mathbf{W}_{\mathrm{uk}}\|)^{-1}
\tag{19e}
$$

$$
\eta_{\mathrm{kr}} = \tau (\|\mathbf{W}_{\mathrm{qr}}\| \|\mathbf{W}_{\mathrm{dq}}\|)^{-1}.
\tag{19f}
$$

We then substitute these learning rates into Eqs. (18), to see that the bounds are given by,

$$
\begin{aligned}
|(\Delta q_{\mathrm{nope}})^T k_{\mathrm{nope}}| &\leq \tau D + \tau D + O \left( \frac{\tau^2 \|\mathbf{W}_{\mathrm{uk}}\| \|\mathbf{W}_{\mathrm{dkv}}\|}{\|\mathbf{W}_{\mathrm{dq}}\| \|\mathbf{W}_{\mathrm{uk}}\| \|\mathbf{W}_{\mathrm{dkv}}\| \cdot \|\mathbf{W}_{\mathrm{uq}}\| \|\mathbf{W}_{\mathrm{uk}}\| \|\mathbf{W}_{\mathrm{dkv}}\|} \right) \\
&= 2\tau D + O \left( \frac{\tau^2}{\|\mathbf{W}_{\mathrm{dq}}\| \|\mathbf{W}_{\mathrm{uq}}\| \|\mathbf{W}_{\mathrm{uk}}\| \|\mathbf{W}_{\mathrm{dkv}}\|} \right),
\end{aligned}
\tag{20a}
$$

$$
\begin{aligned}
|(\Delta q_{\mathrm{rope}})^T k_{\mathrm{rope}}| &\leq \tau D + \tau D + O \left( \frac{\tau^2 \|\mathbf{W}_{\mathrm{kr}}\|}{\|\mathbf{W}_{\mathrm{dq}}\| \|\mathbf{W}_{\mathrm{kr}}\| \cdot \|\mathbf{W}_{\mathrm{qr}}\| \|\mathbf{W}_{\mathrm{kr}}\|} \right) \\
&= 2\tau D + O \left( \frac{\tau^2}{\|\mathbf{W}_{\mathrm{dq}}\| \|\mathbf{W}_{\mathrm{qr}}\| \|\mathbf{W}_{\mathrm{kr}}\|} \right),
\end{aligned}
\tag{20b}
$$

$$
\begin{aligned}
|q_{\mathrm{nope}}^T \Delta k_{\mathrm{nope}}| &\leq \tau D + \tau D + O \left( \frac{\tau^2 \|\mathbf{W}_{\mathrm{uq}}\| \|\mathbf{W}_{\mathrm{dq}}\|}{\|\mathbf{W}_{\mathrm{uq}}\| \|\mathbf{W}_{\mathrm{dq}}\| \|\mathbf{W}_{\mathrm{dkv}}\| \cdot \|\mathbf{W}_{\mathrm{uq}}\| \|\mathbf{W}_{\mathrm{dq}}\| \|\mathbf{W}_{\mathrm{uk}}\|} \right) \\
&= 2\tau D + O \left( \frac{\tau^2}{\|\mathbf{W}_{\mathrm{uq}}\| \|\mathbf{W}_{\mathrm{dq}}\| \|\mathbf{W}_{\mathrm{uk}}\| \|\mathbf{W}_{\mathrm{dkv}}\|} \right),
\end{aligned}
\tag{20c}
$$

$$
|q_{\mathrm{rope}}^T \Delta k_{\mathrm{rope}}| \leq \tau D.
\tag{20d}
$$

It is reasonable to assume in practice that the weights are not arbitrarily small (i.e. their norm is lower bounded), and thus that these terms are bounded by a constant.

The only remaining term to bound in Eq. (15) is the quadratic term, $\|\Delta q\|\|\Delta k\|$. We can show that this is bounded by showing that the individual parts are bounded,

$$\|\Delta q_{\text{nope}}\| \leq \eta_{\text{uq}} D\|\mathbf{W}_{\text{dq}}\| + \eta_{\text{dq}} D\|\mathbf{W}_{\text{uq}}\| + \eta_{\text{uq}}\eta_{\text{dq}} D^2$$

$$\leq \frac{2\tau D}{\|\mathbf{W}_{\text{uk}}\|\|\mathbf{W}_{\text{dkv}}\|} + \eta_{\text{uq}}\eta_{\text{dq}} D^2, \tag{21a}$$

$$\|\Delta q_{\text{rope}}\| \leq \eta_{\text{qr}} D\|\mathbf{W}_{\text{dq}}\| + \eta_{\text{dq}} D\|\mathbf{W}_{\text{qr}}\| + \eta_{\text{qr}}\eta_{\text{dq}} D^2$$

$$\leq \frac{2\tau D}{\|\mathbf{W}_{\text{kr}}\|} + \eta_{\text{qr}}\eta_{\text{dq}} D^2, \tag{21b}$$

$$\|\Delta k_{\text{nope}}\| \leq \eta_{\text{uk}} D\|\mathbf{W}_{\text{dkv}}\| + \eta_{\text{dkv}} D\|\mathbf{W}_{\text{uk}}\| + \eta_{\text{uk}}\eta_{\text{dkv}} D^2$$

$$\leq \frac{2\tau D}{\|\mathbf{W}_{\text{uq}}\|\|\mathbf{W}_{\text{dq}}\|} + \eta_{\text{uk}}\eta_{\text{dkv}} D^2, \tag{21c}$$

$$\|\Delta k_{\text{rope}}\| \leq \eta_{\text{kr}} D \leq \frac{\tau D}{\|\mathbf{W}_{\text{qr}}\|\|\mathbf{W}_{\text{dq}}\|}. \tag{21d}$$

To extend further to the multi-head setting, we add head indices to the necessary matrices, $\mathbf{W}_{\text{uq}}^h$, $\mathbf{W}_{\text{uk}}^h$, and $\mathbf{W}_{\text{qr}}^h$, and their corresponding learning rates, $\eta_{\text{uq}}^h$, $\eta_{\text{uk}}^h$, $\eta_{\text{qr}}^h$. The key used for RoPE, $k_{\text{rope}}$ is shared between all heads, therefore $\mathbf{W}_{\text{kr}}$ surprisingly does not have a head index. The down matrices project to a latent space, so also do not have head indices. Plugging these into Eqs. (19) we have,

$$\eta_{\text{uq}}^h = \tau(\|\mathbf{W}_{\text{dq}}\|\|\mathbf{W}_{\text{uk}}^h\|\|\mathbf{W}_{\text{dkv}}\|)^{-1} \tag{22a}$$

$$\eta_{\text{dq}} = \tau \min\left\{ \left(\max_h \|\mathbf{W}_{\text{uq}}^h\|\|\mathbf{W}_{\text{uk}}^h\|\|\mathbf{W}_{\text{dkv}}\|\right)^{-1}, \left(\max_h \|\mathbf{W}_{\text{qr}}^h\|\|\mathbf{W}_{\text{kr}}\|\right)^{-1}\right\} \tag{22b}$$

$$\eta_{\text{qr}}^h = \tau(\|\mathbf{W}_{\text{dq}}\|\|\mathbf{W}_{\text{kr}}\|)^{-1} \tag{22c}$$

$$\eta_{\text{uk}}^h = \tau(\|\mathbf{W}_{\text{uq}}^h\|\|\mathbf{W}_{\text{dq}}\|\|\mathbf{W}_{\text{dkv}}\|)^{-1} \tag{22d}$$

$$\eta_{\text{dkv}} = \tau(\max_h \|\mathbf{W}_{\text{uq}}^h\|\|\mathbf{W}_{\text{dq}}\|\|\mathbf{W}_{\text{uk}}^h\|)^{-1} \tag{22e}$$

$$\eta_{\text{kr}} = \tau(\max_h \|\mathbf{W}_{\text{qr}}^h\|\|\mathbf{W}_{\text{dq}}\|)^{-1}. \tag{22f}$$

The use of $\max_h(\cdot)$ comes from the requirement that we want logit changes to be bounded for all heads.

