# OpenReview forum: "Controlling changes to attention logits"
_ICLR.cc/2026/Workshop/Sci4DL — Submitted to Sci4DL 2026_

### Official Review · Reviewer_BKSr · 2026-02-20

**Fit:** 3
**Significance:** 2
**Confidence:** 2

**Summary:**

This paper proposes a stabilization method for transformer pretraining that controls changes in attention logits rather than their absolute magnitude. The authors derive parameter-dependent learning rates for query and key weights that scale inversely with the norms of the corresponding counterpart matrices, bounding worst-case logit change. In ~1B-sized experiments with MHA and MLA, the proposed method enables higher base learning rates, matches QK norm stability in MHA, and outperforms QK clip in MLA.

**Strengths:**

- The stabilization strategy that controls updates to attention logits rather than their absolute values is intuitive and well motivated.

- The method is lightweight and easy to integrate, requiring only learning-rate adjustments without modifying the attention computation or inference path. This makes it particularly attractive for MLA, where QK normalization is not feasible.

- Empirically, QuacK stabilizes training at high learning rates and performs competitively with QK norm in MHA, while outperforming QK clip in MLA. The MLA extension is carefully derived and practically relevant.

**Suggestions:**

- The theoretical guarantees are limited to first-order, worst-case bounds and rely on strong assumptions. The connection between bounded logit change and overall training dynamics or generalization remains indirect.

- Experiments are restricted to ~1B models and relatively short training runs, leaving open questions about behavior at larger scales and longer horizons.

---

### Official Review · Reviewer_wfNM · 2026-02-24

**Fit:** 1
**Significance:** 1
**Confidence:** 2

**Summary:**

This paper addresses  training instability, by hypothesizing that the instability is mostly driven by large changes in attention logits, rather than their magnitude. They introduce a method to address this, QuacK, and compare across increasing learning rates for QuacK and alternative methods, in the conventional Multi Head Attention (MHA) setting, and in the Multi head Latent Attention (MLA) setting. The paper finds that QuacK leads to stable training across learning rates and better performance in the validation loss in the MLA setting, where QK norm isn't competitive at inference.

In my judgement, these findings aren't properly contextualized with the available results shown in the appendix. As elaborated in the suggestions, Quack doesn't succeed in controlling large logit changes better than alternative baselines, and the chosen baselines don't tease out relevant aspects to validate the proposed hypothesis. These important points aren't mentioned or addressed anywhere in the paper. Therefore, the primary contribution of QuacK relies on its better performance for higher learning rates for MLA. Here too, there is a lingering uncertainty on whether this better performance is valid only up to a given training step, potentially limiting the benefits from this method. Given these shortcomings, I deem the first two stated contributions of the paper to be incorrect and/or incomplete, and the remaining contributions seem incomplete and insufficient at this stage.

**Strengths:**

The paper
* identifies a real and current problem: a lack of performant methods to address training instability for the Multi-head Latent Attention setting.
* proposes an interesting new method to address it, with principled motivation based on the gradient structure and empirical motivation to apply their method in the Multi head Latent Attention setting, where other methods, such as QK Norm, aren't competitive due to inference costs.
* compares to multiple baselines, including alternative proposed methods and two informative baselines: an ablation baseline (where fixed QK learning rates are set across training) and the default setting (only in the Appendix.)

**Suggestions:**

Feedback on methodology and analysis

* Evaluation of hypothesis and choice of baseline:
	* The hypothesis distinguishes between the absolute magnitude of the logit activations and the magnitude of the changes to these logits. None of the chosen baselines manages to tease out the distinction between these two cases, as shown in Fig. 3 of the Appendix, as they seem to be fairly correlated, ie metrics have the same relative scaling across both metrics. Therefore, training performance stability could be just as well attributed to the absolute magnitude, instead of logit changes.  It would be valuable to construct a method that has low logit changes, but high absolute magnitude. In this setup, they could verify whether their hypothesis is correct.
	* The hypothesis isn't evaluated after discussing the results (Fig. 2), whereas Fig. 3 is not discussed at all, either in the main body or in the Appendix.
	* Since the hypothesis isn't validated by the experiments shown in the paper, it follows that the first stated contribution of the paper also isn't validated (in Introduction).

* Validation of QuacK and comparison to baselines.
	* The theoretical motivation for QuacK seems to not be empirically validated when analyzing Fig. 3, where QK Norm and the Ablation baselines lead to both lower max. logits and lower avg. logit score.
	* This is important to contextualize the significance of the second contribution of the paper (in Introduction). Even though QuacK succeeds in controlling logit changes, it doesn't do so better than the ablation baseline, which uses a fixed learning rate, or other alternative methods, such as QK Norm. This disfavors the QuacK method, and by extention, the second contribution.
	* This judgement is based on the results of Fig. 3 being valid across heads and layers, which is not clearly established by the results in the paper and Appendix.
	* It would have been nice to see:
		* Fig. 3 (in appendix) addressed and discussed in the main body, or at least discussed in the Appendix. It was great to see that this was included in the paper, but it is not sufficient to only include it in the Appendix without any discussion or mention, since it is necessary to evaluate the main method proposed by the paper.
		* A more extensive (across layers and heads) and thorough analysis along the lines of the metrics in Fig. 3. I didn't find it mentioned anywhere in the paper or Appendix that Fig. 3 is representative. It might be that this extended analysis reveals a different picture.

* Ablation versus QuacK performance:
	* In Fig. 2 for MLA and eta=3e-3, the Ablation baseline validation loss seems to converge towards QuacK at the end of training. Is this a common observation across learning rates, if one trains longer? Is it such that QuacK only outperforms Ablation only above some training step threshold that is dependent on the learning rate?
	* This observation is not discussed in the paper, and it is important for evaluating the third stated contribution of the paper (see Introduction).

---

### Official Review · Reviewer_gHgc · 2026-03-02

**Fit:** 2
**Significance:** 2
**Confidence:** 2

**Summary:**

The paper identifies unstable attention logit changes as a primary driver of training instability in transformers, particularly when using high learning rates. The authors propose QuacK, a method that stabilizes training by dynamically modulating the learning rates of query and key weight matrices based on each other's norms.

**Strengths:**

The motivation and method is more or less clearly presented. One strength of the proposed stabilization is unlike QK normalization, QuacK is compatible with Multi-head Latent Attention (MLA), and seems to improve upon QK clip at large LR.

**Suggestions:**

A few questions that might be useful for the authors to consider

1. QuacK is described as a 'cheap' alternative to QK norm, which adds a normalization layer to the forward pass. Have you conducted a  forward-pass latency comparison or measured the tokens-per-second throughput to quantify exactly how much overhead QK norm adds relative to your method?

2. The experiments are presented when training with Muon, this is an obvious step that the authors will have to try training with various different adaptive optimizers to see how their stabilization behaves. Different model sizes, datasets are also other obvious things needed to make it more thorough.

3. With $\mu$P we see relatively fast zero-shot hyperparameter transfer. It would be good to demonstrate if the optimal $\tau$ (relative learning rate) found on smaller models remains optimal for the 1B+ scale.

---

### Meta-Review · Area_Chair_R4S4 · 2026-03-01

**Recommendation:** Reject

**Metareview:**

The claims made in the paper, while theoretically motivated, are not backed by the experimental results as they do not disentangle the absolute magnitude of the logit activations and the magnitude of the changes to these logits effectively, and the proposed method does not perform significantly better than baselines (in the sense that it is hard to know if the difference is statistically significant). I suggest the authors review the reviewers' feedback and take it into account.

---

### Decision · Program_Chairs · 2026-03-02

Reject